# Single-Phase Inverter Deadbeat Control with One-Carrier-Period Lag

**Wei Yao** [1] **, Jiamin Cui** [1] **and Wenxi Yao** [2,*]

[1] Key Laboratory for Technology in Rural Water management of Zhejiang Province, College of Electrical Engineering, Zhejiang University of Water Resources and Electric Power, Hangzhou 310018, China; yaowei@zjweu.edu.cn (W.Y.); cuijm@zjweu.edu.cn (J.C.)
[2] College of Electrical Engineering, Zhejiang University, Hangzhou 310027, China
* Correspondence: ywxi@zju.edu.cn; Tel.: +86-135-8829-3953

**Abstract:** This paper presents a novel digital control scheme for the regulation of single-phase voltage source pulse width modulation (PWM) inverters used in AC power sources. The proposed scheme adopts two deadbeat controllers to regulate the inner current loop and the outer voltage loop of the PWM inverter. For the overhead of digital processing, the change of duty of PWM lags one carrier period behind the sampling signal, which is modeled as a first-order lag unit in a discrete domain. Based on this precise modeling, the deadbeat controllers make the inverter get a fast dynamic response, so that the inverter's output voltage is obtained with a very low total harmonic distortion (THD), even when the load is fluctuating. The parameter sensitivity of the deadbeat control was analyzed, which shows that the proposed deadbeat control system can operate stably when the LC filter's parameters vary within the range allowed. The experimental results of a 2kW inverter prototype show that the THD of the output voltage is less than 3% under resistive and rectifier loads, which verifies the feasibility of the proposed scheme. An additional advantage of the proposed scheme is that the parameter design of the controller can be fully programmed without the experience of a designer.

**Keywords:** inverter; deadbeat control; digital control; THD; DSP (digital signal processor)

## 1. Introduction

DSP-based pulse width modulation (PWM) inverters with LC filters have found widespread application in AC power conditioning systems, such as uninterruptible power supplies (UPS), programmable AC power sources, and green energy converters. Digital control of PWM inverters has become the mainstream control technology. Compared with analog control, digital inverters have well-known advantages, but also bring some problems [1–3]. One of these is that the PWM duty cycle cannot reach full range from 0 to 1 due to the overhead of digital processing, which results in a loss of output voltage. To avoid this problem, one-PWM-period control lag is implemented; that is, after sampling and calculation, the output of the current control value is not immediate, but occurs only when the next carrier period of PWM begins. However, the control lag may affect the inverter's stability. In the discrete realization of the Proportional-Integral-Differential (PID) control method, it is tough work to design a digital controller that meets the demands of both rapidity and stability, because the control bandwidth of inverters is not only limited by the PWM switching frequency but also by the control lag. Wider control bandwidth leads to better dynamic control performance. In addition to the classical control method, repetitive control [4–6] and proportional-resonant control [7–9] are employed in inverters, which feature perfect steady-state performance but degraded dynamic performance. Deadbeat control is a digital control method aimed at the fastest elimination of tracking errors. In order

to obtain high-quality sinusoidal output voltage under various loads, many researchers have focused on the study of the deadbeat control of inverters. Deadbeat control designed directly in a discrete domain can be more accurate and precise than PID control. In [10,11], a deadbeat control method based on a single-voltage-loop inverter was proposed. In [12,13], the deadbeat control of multi-loop inverters was proposed, which was implemented without the one-beat lag of the digital control. The range of output voltage was limited in this scheme. Excluding this fault, the multi-loop structure could achieve better inverter performance than the single-loop structure. In [14–17], predictive control was added to offset the lag of the control, so there was no duty limit in the deadbeat control of the inverter. Theoretically, the output of this system can trace the input in one beat. However, the actual system is very sensitive to parameter drift and load disturbance, which may even lead to instability.

This paper proposes a deadbeat control scheme for multi-loop inverters with a one-carrier-period control lag. In a discrete domain, the one-carrier-period control lag is modeled as a first-order lag unit, i.e., $z^{-1}$, which is combined with the discrete controlled object as a whole plant, and the unit-step signal is regarded as the input. Then the z-transfer function of the controller is derived according to a deadbeat control algorithm. The stability analysis of the proposed scheme is given in an allowable range of parameter drift. The simulation and experiment results verify the feasibility of this scheme. Excellent voltage waveform quality is obtained, which proves the perfect steady-state and dynamic-state performance of the proposed scheme.

## 2. Deadbeat Control Design of Multi-Loop Inverter Control Structure

### 2.1. Multi-Loop Inverter Control Structure

Figure 1 shows a PWM inverter system. The full-bridge inverter is connected to the load through LC filtering. The voltage source, $V_{dc}$, serves as the DC bus and supplies power to the load through the full-bridge inverter. The PWM inverter illustrated in Figure 1 is a nonlinear system by nature due to the existence of the switching devices. State-space averaging and linearization techniques can be applied to model the switching converters as linear systems. The control goal is to make the output voltage, $v_o$, trace the sinusoidal reference voltage, $v_{ref}$, as quickly as possible, while maintaining stability. Load current, $i_o$, is treated as disturbance in this case.

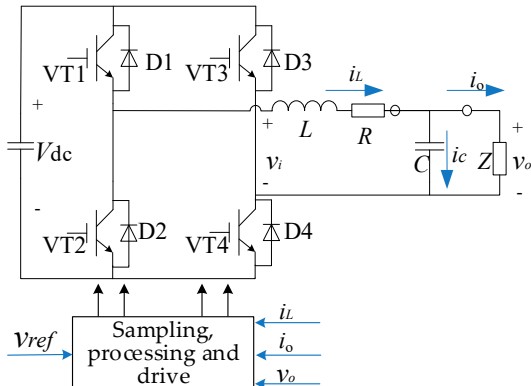

**Figure 1.** Pulse width modulation (PWM) inverter system.

The capacitance voltage, $v_o$, and inductance current, $i_L$, are taken as state variables. The dynamic equations of the PWM inverter can be derived as

$$\begin{cases} C\frac{dv_o}{dt} = i_L - i_o \\ L\frac{di_L}{dt} = -ri_L - v_o + v_i \end{cases} \tag{1}$$

where $v_i$ is the output voltage of the full bridge and $R$ is the equivalent resistor of the inductor.

Figure 2 shows a diagram of the multi-loop inverter control structure [11–13], in which the model of Plant corresponds to Equation (1). Three signals are sensed: the inductor current, $i_L$, the output voltage, $v_o$, and the load current, $i_o$. The inductor current is sensed for the regulation of the current loop. The output voltage is sensed for both the voltage regulation and the "back electromotive force (EMF)" decoupling. The load current is sensed for the feedforward to compensate the load fluctuation. Through the two feedforward channels, the outer-loop voltage has no effect on the inner-loop current and the load current has little effect on the output voltage. Then the multi-loop control structure is simplified to the dual-loop control structure shown in Figure 3.

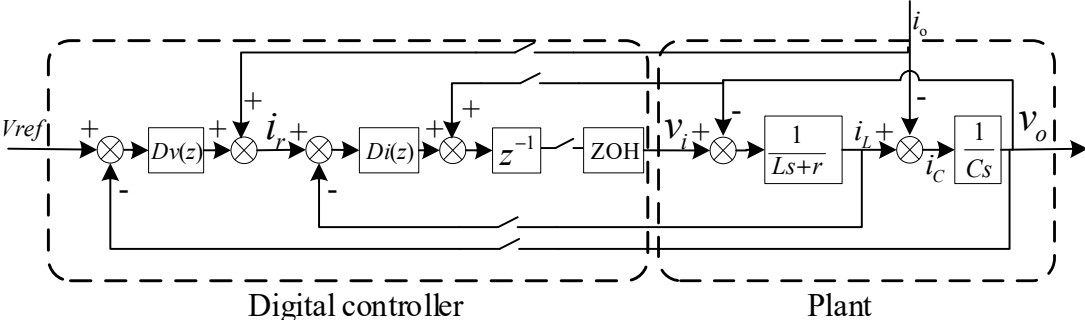

**Figure 2.** Multi-loop inverter control structure.

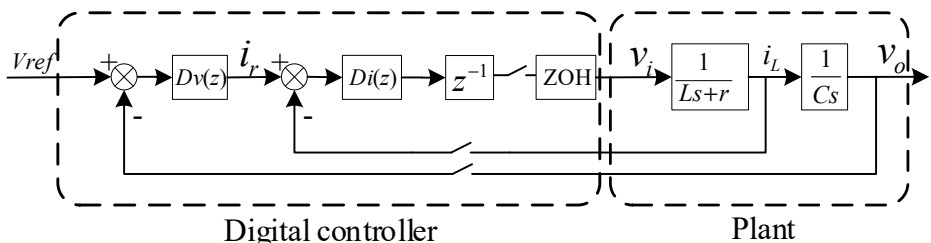

**Figure 3.** Simplified dual-loop control structure.

In a discrete domain, the one-beat lag of the control is modeled as $z^{-1}$, which is taken apart from the controller and combined with the actual plant. Then the controller can be designed according to deadbeat control theory. Firstly, an inner current loop controller is designed, then an outer voltage controller is designed.

## 2.2. Deadbeat Control Theory

Deadbeat control is used to make closed-loop systems achieve a steady state without error for a specific input in a minimum beat. Deadbeat control is one of the digital control schemes, and a sampling period is called one beat. The closed-loop z-transfer function of deadbeat control can be described as

$$\Phi(z) = m_1 z^{-1} + m_2 z^{-2} + \cdots + m_N z^{-N} \tag{2}$$

where $N$ is the smallest integer possible and $m_1 \cdots m_N$ are the undetermined coefficients of the polynomials. This z-transfer function shows that the impulse response of a closed-loop system reaches zero after $N$ sampling periods. This means the system reaches a stable state after $N$-beat regulation. If $N$ is the minimum value possible, this control is called deadbeat control, which can achieve the fastest dynamic response, in theory. Providing the discrete controlled object transfer function is known to be $G(z)$, according to classical control theory the discrete closed-loop transfer function can be described as

$$\Phi(z) = \frac{D(z)G(z)}{1 + D(z)G(z)} \tag{3}$$

If the value of $N$ and the coefficients $m_1 \cdots m_N$ in Equation (2) were determined, the deadbeat controller z-transfer function could be derived as

$$D(z) = \frac{1}{G(z)} \frac{\Phi(z)}{1 - \Phi(z)} \tag{4}$$

The design of $D(z)$ must also consider the stability and realizable constraints of the controller. If there is no time-delay element in the control loop, the fastest regulation can reach a steady state in one beat. If there are $k$ beats of the time-delay block in the control loop, the minimum number of beats of regulation must be greater than $k$. $D(z)$ and $G(z)$ appear in pairs, but their unstable poles and zeros on and outside the unit circle do not allow each other to be cancelled out. If the parameters of the controlled object drift or the identification parameters have errors, this cancellation cannot be realized accurately, thus causing instability.

The design of the deadbeat controller of the inverter is described in terms of the inner current loop and the outer voltage loop, respectively, in the following section.

### 2.2.1. Design of the Deadbeat Controller of the Inner Current Loop

The multi-loop control structure of the inverter is simplified into a dual-loop structure in which the current loop is extracted, as shown in Figure 4.

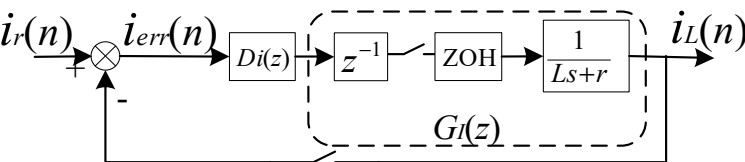

**Figure 4.** Inverter current loop.

In Figure 4, the block ZOH denotes a zero-order holder and has its s-transfer function as

$$G_h(s) = \frac{1 - e^{-Ts}}{s} \tag{5}$$

The series model of the zero-order holder and the inductor admittance is discretized as

$$G_a(z) = Z\left( \frac{1 - e^{-Ts}}{s} \cdot \frac{1}{Ls + r} \right) = \frac{1 - e^{-\frac{r}{L}T}}{r} \cdot \frac{z^{-1}}{1 - e^{-\frac{r}{L}T} \cdot z^{-1}} \tag{6}$$

where $Z(*)$ means z-transform and $T$ is sampling time. Considering the one-beat latency of the control as $z^{-1}$, the discrete controlled object is obtained as

$$G_I(z) = \frac{1 - e^{-\frac{r}{L}T}}{r} \cdot z^{-1} \cdot \frac{z^{-1}}{1 - e^{-\frac{r}{L}T} \cdot z^{-1}} \tag{7}$$

providing that the input of the closed-loop system is a step command, which is described in the Z domain as

$$R(z) = \frac{1}{1 - z^{-1}} \tag{8}$$

The z-transform of the error sequence of the current loop is described as $R(z) \cdot (1 - \Phi_I(z))$, where $\Phi_I(z)$ is the current closed-loop z-transfer function with the form of Equation (2). According to the final value theorem of z-transform, the current error sequence in a steady state is

$$\lim_{n \to \infty} i_{err}(n) = \lim_{z \to 1}\left(1 - z^{-1}\right) \cdot R(z) \cdot (1 - \Phi_I(z)) = \lim_{z \to 1}\left(1 - z^{-1}\right) \cdot \frac{1}{1 - z^{-1}} \cdot (1 - \Phi_I(z)) = \lim_{z \to 1}(1 - \Phi_I(z)) \tag{9}$$

In order to make zero error in a steady state, polynomial $(1 - \Phi_I(z))$ must contain the factor $(1 - z^{-1})$. Therefore, polynomial $(1 - \Phi_I(z))$ can be written in the form

$$1 - \Phi_I(z) = \left(1 - z^{-1}\right) f\!\left(z^{-1}\right) \tag{10}$$

where $f\!\left(z^{-1}\right)$ is the undetermined polynomial of $z^{-1}$.

From Equation (2) of the deadbeat controller, polynomial $\Phi_I(z)$ must contain the factor $z^{-1}$. If factor $z^{-1}$ did not emerge in polynomial $\Phi_I(z)$, it should emerge in polynomial $D_I(z)$ in the form of $z^{+1}$. Polynomial $D_I(z)$ containing the factor $z^{+1}$ is noncausal and makes the controller impossible. Therefore, polynomial $\Phi_I(z)$ can be written in the form

$$\Phi_I(z) = z^{-1}(m_1 z^{-1} + m_2 z^{-2} + \cdots + m_N z^{-N}) \tag{11}$$

When $N = 1$ and $m_1 = 1$, deadbeat control is achieved.
Therefore,

$$\Phi_I(z) = z^{-2} \tag{12}$$

And

$$1 - \Phi_I(z) = 1 - z^{-2} = \left(1 - z^{-1}\right)\left(1 + z^{-1}\right) \tag{13}$$

which is consistent with Equation (10) and meets the requirement of zero error in a steady state. According to Equations (4), (12), and (13), the controller z-transfer function could be obtained as

$$D_I(z) = \frac{r}{1 - e^{-\frac{rT}{L}}} \cdot \frac{1 - e^{-\frac{rT}{L}} \cdot z^{-1}}{1 - z^{-2}} \tag{14}$$

### 2.2.2. Design of the Deadbeat Controller of the Outer Voltage Loop

The designed inner loop can follow the current command in two beats modeled as $z^{-2}$. Therefore, the voltage loop is simplified as in Figure 5.

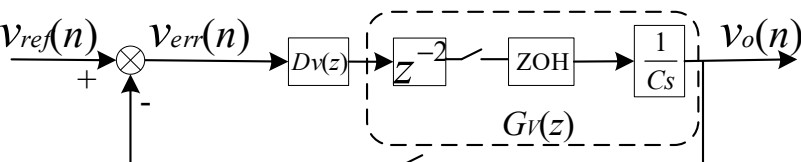

**Figure 5.** Inverter voltage loop.

The discrete controlled object of the voltage loop can be obtained as

$$G_V(z) = z^{-2} \cdot Z\!\left(\frac{1 - e^{-Ts}}{s} \cdot \frac{1}{Cs}\right) = \frac{T}{C} \cdot z^{-2} \cdot \frac{z^{-1}}{1 - z^{-1}} \tag{15}$$

Providing that the input of the closed-loop system is a unit-step command, the z-transform of the error sequence of the voltage loop is described as $R(z) \cdot (1 - \Phi_V(z))$, where $\Phi_V(z)$ is the voltage closed-loop z-transfer function with the form of Equation (2). According to the final value theorem of z-transform, the voltage error sequence in a steady state is

$$\lim_{n \to \infty} u_{err}(n) = \lim_{z \to 1}\left(1 - z^{-1}\right) \cdot R(z) \cdot (1 - \Phi_V(z)) = \lim_{z \to 1}\left(1 - z^{-1}\right) \cdot \frac{1}{1 - z^{-1}} \cdot (1 - \Phi_V(z)) = \lim_{z \to 1}(1 - \Phi_V(z)) \tag{16}$$

In order to make zero error in a steady state, polynomial $(1 - \Phi_V(z))$ must contain the factor of $(1 - z^{-1})$. Therefore, polynomial $(1 - \Phi_V(z))$ can be written in the form

$$1 - \Phi_V(z) = \left(1 - z^{-1}\right)f\left(z^{-1}\right) \tag{17}$$

where $f\left(z^{-1}\right)$ is the undetermined polynomial of $z^{-1}$. Similar to the analysis of the current loop, polynomial $\Phi_V(z)$ can be written in the form

$$\Phi_V(z) = z^{-2}\left(m_1 z^{-1} + m_2 z^{-2} + \cdots + m_N z^{-N}\right) \tag{18}$$

When $N = 1$ and $m_1 = 1$, deadbeat control is achieved.

Therefore,

$$\Phi_V(z) = z^{-3} \tag{19}$$

And

$$1 - \Phi_V(z) = 1 - z^{-3} = \left(1 - z^{-1}\right)\left(1 + z^{-1} + z^{-2}\right) \tag{20}$$

which is consistent with Equation (10) and meets the requirement of zero error in a steady state. According to Equations (4), (12) and (13), the controller z-transfer function could be obtained as

$$D_V(z) = \frac{C}{T} \cdot \frac{1}{1 + z^{-1} + z^{-2}} \tag{21}$$

## 3. Simulation and Experiment

The proposed deadbeat control with a one-carrier-period control lag was verified through simulation and experiment. The design parameters of the inverter are listed in Table 1.

**Table 1.** Design parameters of the inverter.

| | |
|---|---|
| **Rated power** | 2.4 kW |
| **Rated output voltage** | 220 V |
| **DC bus voltage** | 400 V |
| **Filter inductance** | 1.2 mH |
| **Inductance resistance** | 0.68 Ω |
| **Filter capacitance** | 30 uF |
| **PWM frequency** | 16 kHz |
| **Sampling frequency** | 16 kHz |

From Equation (14) and Table 1, the z-transfer function of the current controller is obtained as

$$D_I(z) = \frac{19.54 - 18.86 z^{-1}}{1 - z^{-2}} \tag{22}$$

From Equation (21) and Table 1, the z-transfer function of the voltage controller is obtained as

$$D_V(z) = \frac{0.48}{1 + z^{-1} + z^{-2}} \tag{23}$$

### 3.1. Robustness of the Proposed Control Scheme

From Equation (2), the poles of the closed-loop deadbeat control system were located at the zero point in the complex plane. If the system parameters drift, the actual control will deviate from the expected, and even the system will be unstable. In this study, the inductive core was made from

iron-nickel high-flux magnetic powder, and the inductance decreased with the increase of the current. Resistance had a positive temperature characteristic. According to the data sheet and the experimental data, the variation of the inductance was within 0.6–1 times of the design value and the variation of resistance was within 1–1.7 times of the design value. Figure 6 shows the pole distribution of the current loop with the drift of the inductance and resistance. Considering that the variation of the capacitance was within 0.7–1.1 times of the nominal value, the pole distribution of the voltage loop could be plotted, as shown in Figure 7. In Figures 6 and 7, it can be seen that the poles are distributed in a unit circle centered at the origin, guaranteeing the robustness of the proposed scheme.

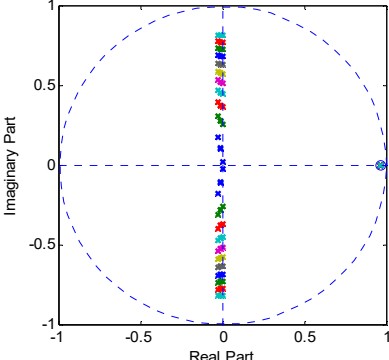

**Figure 6.** Pole distribution of the current loop.

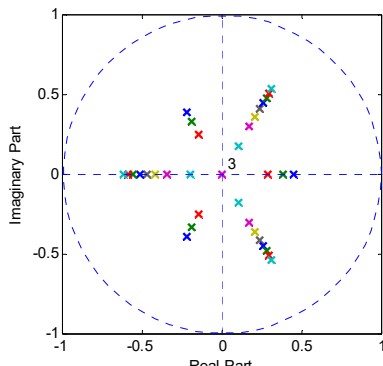

**Figure 7.** Pole distribution of the voltage loop.

The reference voltage in the proposed control scheme was a sampled sinusoidal wave which was constant in one PWM carrier period. The reference voltage could be regarded as the time shift superposition of a series of unit-step signals. Using state-space averaging, the model of the inverter belongs to the linear system which satisfies the superposition theorem. Therefore, it is reasonable to take the unit-step signal as the input signal. The same analysis applies to the reference current. The following simulation experiments verify these analyses.

### 3.2. Simulation Results

Referring to Figure 2, a simulation model was set up, and the simulation results of the output voltage and the current are shown in Figure 8. Figure 8a shows the simulation results of the output voltage and the current under resistive load variation from half load to full load. Figure 8b shows the simulation results of the output voltage and the current under rectifier load. From these results, it follows that the controlled PWM inverter can maintain a sinusoidal waveform even under a sudden load change and periodic fluctuation.

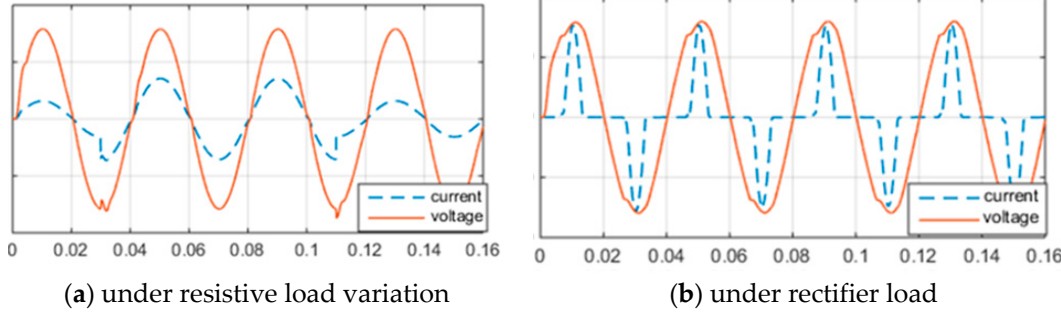

(**a**) under resistive load variation          (**b**) under rectifier load

**Figure 8.** Simulation waveform output voltage and current.

A conventional PID controller was also designed for comparison with the deadbeat controller. By means of classical design and engineering optimization [13,18,19], the PI parameters of the dual-loop controller were obtained. The output voltage total harmonic distortion (THD) comparison of the two control schemes is shown in Table 2. From the data, it can be seen that the quality of the output voltage of the proposed scheme was obviously better than that of the conventional PI scheme.

**Table 2.** Output voltage total harmonic distortion (THD) comparison between the PI scheme and the proposed scheme.

| | Resistive Load | | | Rectifier Load | | |
|---|---|---|---|---|---|---|
| | **Full Load** | **Half Load** | **Empty Load** | **Full Load** | **Half Load** | **Empty Load** |
| The PI scheme | 2.02 | 1.83 | 0.87 | 2.63 | 2.37 | 1.52 |
| The proposed | 1.62 | 1.39 | 0.38 | 2.34 | 2.11 | 1.27 |

### 3.3. Experiment Results

A DSP-controlled PWM inverter was implemented to verify the proposed digital control scheme. The output voltage, inductor current, and load current were sensed as feedback for the purpose of closed-loop regulation. A digital signal processor, TMS320F28035, from Texas Instruments Inc. was chosen as the main processor of the digital control system. The reference command in the proposed control scheme was a sampled sinusoidal wave stored in the program memory of the DSP. Four IGBTs were selected for the power switch, and the model number was FGH60N60.

Figure 9 shows the experimental results of the output voltage and current under a resistive load switching from half load to full load. The voltage drop at the instance of load variation could be quickly recovered by the designed digital controller. Figure 10 depicts the experimental waveform under a rectifier load with the current crest factor of 3.4. The THD of the output voltage was 2.96% in this case. Although a highly nonlinear load was connected, the sinusoidal output waveform could still be maintained. Full load range experiments under resistive and rectifier loads were carried out, and Table 3 shows the experimental data of the output voltage THD and RMS (Root Mean Square) under resistive and rectifier loads. In the full load range, the THD was less than 3% and the RMS was very close to the reference. It can be seen from these results that the proposed multiple-loop scheme worked well even under rough load conditions.

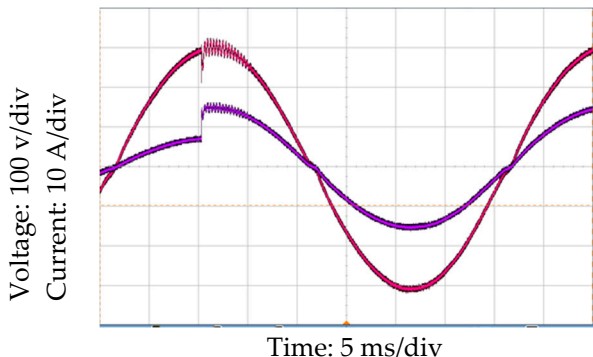

**Figure 9.** Experimental results of the output voltage and the load current.

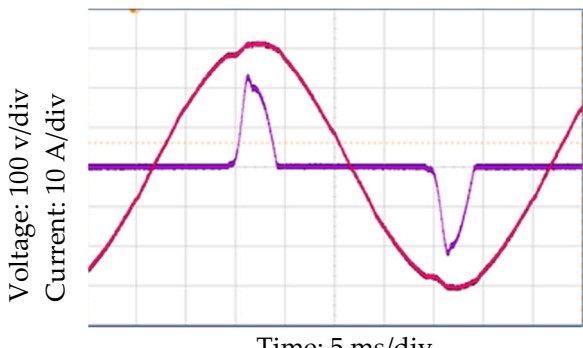

**Figure 10.** Experimental results of the output voltage and the load current.

**Table 3.** Experimental data of the output voltage under resistive and rectifier loads.

| | Resistive Load | | | Rectifier Load | | |
|---|---|---|---|---|---|---|
| | **Full Load** | **Half Load** | **Empty Load** | **Full Load** | **Half Load** | **Empty Load** |
| THD (%) | 1.90 | 1.84 | 0.98 | 2.96 | 2.61 | 1.87 |
| RMS (V) | 216 | 212 | 214 | 215 | 215 | 213 |

Resistive load: full load is 20 Ω and half load is 40 Ω. Rectifier load: full load is 50 Ω, half load is 100 Ω, and capacitance is 3300 uF.

## 4. Conclusions

In this paper, we have presented a single-phase inverter deadbeat control scheme with one-carrier-period control lag. Through accurate modeling, the proposed scheme can obtain good static and dynamic performance. A digital PWM inverter control system based on DSP TMS320F28035 was constructed to verify the proposed control scheme. The experimental results show that the controlled PWM inverter can sustain a large load variation and an output voltage of less than 3% THD in the full load range. This indicates that the proposed scheme is promising for the application of single-phase inverters. The proposed scheme can maximize the control bandwidth when the PWM frequency and sampling frequency are limited and control lag exists. For this reason, this scheme is worth trying in high-power applications.

**Author Contributions:** Conceptualization, W.Y. (Wei Yao) and W.Y. (Wenxi Yao); methodology, W.Y. (Wei Yao); software, J.C.; validation, W.Y. (Wei Yao) and J.C.; formal analysis, W.Y. (Wei Yao); investigation, J.C.; resources, W.Y. (Wenxi Yao); data curation, J.C.; writing—original draft preparation, W.Y. (Wei Yao); writing—review and editing, J.C. and W.Y. (Wei Yao); visualization, J.C.; supervision, W.Y. (Wenxi Yao); project administration, W.Y. (Wenxi Yao); funding acquisition, W.Y. (Wenxi Yao). All authors have read and agreed to the published version of the manuscript.

**Funding:** This research was funded by the National Natural Science Foundation of China under grant no. 51677168. This research was also funded by the Research Fund of ZJWEU under grant no. xkyyb201203.

**Conflicts of Interest:** The authors declare no conflict of interest.

## Nomenclature

| | |
|---|---|
| $V_{dc}$ | DC bus voltage |
| $v_{ref}$ | reference voltage |
| $i_o$ | load current |
| $v_o$ | capacitance voltage |
| $i_L$ | inductance current |
| $v_i$ | output voltage of the full bridge |
| $R$ | equivalent resistance of the inductor |
| $L$ | filter inductor |
| $C$ | filter capacitor |
| $\Phi(z)$ | closed-loop z-transfer function of the deadbeat control |
| $D(z)$ | deadbeat controller z-transfer function |
| $G_I(z)$ | z-transfer function of the discrete controlled object in the current loop |
| $R(z)$ | z-transfer function of the unit step |
| $i_{err}(n)$ | current-loop error sequence |
| $\Phi_I(z)$ | z-transfer function of the current closed loop |
| $f(z^{-1})$ | undetermined polynomial of $z^{-1}$ |
| $D_I(z)$ | deadbeat controller z-transfer function of the current loop |
| $G_V(z)$ | z-transfer function of the discrete controlled object in the voltage loop |
| $u_{err}(n)$ | voltage error sequence |
| $\Phi_V(z)$ | z-transfer function of the voltage closed loop |
| $D_V(z)$ | deadbeat controller z-transfer function of the voltage loop |

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
