# Peer review of "Single-Phase Inverter Deadbeat Control with One-Carrier-Period Lag"

_electronics, doi:10.3390/electronics9010154_

Round 1

Reviewer 1 Report

All the listed references should be cited in the text. It seems that the control approach taken in this work is not novel to control engineers. The adavnces or improvements given in this article should be highlighted. Why was the deadbeat control approach taken? what is the sampling period in the experiments?

Author Response

1, All the listed references should be cited in the text.

Answer: I have done it. Pls see the revised manuscript.

2, It seems that the control approach taken in this work is not novel to control engineers.

Answer: Although deadbeat control is not novel in the field of control engineering, In the field of inverter, the classical control is still the mainstream control approach in which controller is designed in continuous domain and then implemented in discrete domain by MCU chip. Deadbeat controller is directly designed in discrete domain, so it’s easy to implemented by MCU. The design of classical controller can only deal with lagging unit approximately in continuous domain. But in discrete domain, lagging unit can be treated accurately. So far, no one has done so detailed research in deadbeat control of inverter. The proposed scheme can achieve higher quality voltage waveform than traditional PI control.

3, The advances or improvements given in this article should be highlighted.

Answer: I have done it. Pls see the revised manuscript.

4, Why was the deadbeat control approach taken?

Answer: Now digital controlled is still the mainstream application in inverter application. Deadbeat control is a digital control method aimed at the fastest elimination of tracking error, which can achieve the best dynamic and static performance in discrete domain. But PID control cannot achieve such performance in theory. In order to achieve better inverter voltage under various loads, I think deadbeat control is better choice.

5, what is the sampling period in the experiments?

Answer: The sampling period is 16kHz same with PWM period.

Reviewer 2 Report

In this paper, the authors proposed a digital control scheme for regulation of single-phase voltage source PWM (pulse width modulation) inverters, where two deadbeat controllers were used to achieve low THD (total-harmonic-distortion). This paper is scientifically sound and contains sufficient interest, I think. However, the research survey of this work is thin. The discussed techniques are out of date. Besides, due to the lack of comparison between the proposed technique and the latest techniques, the effectiveness of this work is not clear.

Reviewer’s comments:

The research survey is not enough. The articles listed in References are old. (Most of them are out of date. The publication date of the newest article is 2015. Five years ago!) You must survey past studies in detail. Besides, you must justify the effectiveness of the proposed technique by comparing with the state-of-the-art techniques.

Which articles did you compare with the proposed technique in Sect.3? The effectiveness of the proposed technique is not clear. Indicate the reference number in sentences. Besides, several techniques have been proposed in past studies. You must justify the effectiveness of the proposed technique by comparing with the latest tecniques.

Don’t use acronym, such as PWM, THD, ZOH, and EMF, without explanation.

In the introduction part, the research survey is thin. The existing techniques discussed in Sect. 1 is out of date. (The authors discussed the techniques proposed more than 10 years ago.) In-depth research survey is necessary. Due to poor research survey, readers will no be able to understand the problem definition of this work.

In the Introduction part, strong points of this proposed method should be further stated and organization of this whole paper is supposed to be provided in the end.

To help readers’ understanding, you should add a notation list.

Unify the expression, such as Gi(z) and GI(z), in figures and equations.

In Sect.3, the simulation and experimental conditions are not clear. What kind of transistors did you use?

In Sect. 3, the authors discussed only the THD characteristics. Other important electrical characteristics, such as power efficiency and power factor, are not clear. Demonstrate the characteristics of the inverter clearly.

Future works as an integral part should be included in the Conclusions.  

Author Response

1, The research survey is not enough. The articles listed in References are old. (Most of them are out of date. The publication date of the newest article is 2015. Five years ago!) You must survey past studies in detail. Besides, you must justify the effectiveness of the proposed technique by comparing with the state-of-the-art techniques.

Answer: I've added that, pls see the revised manuscript.

2, Which articles did you compare with the proposed technique in Sect.3? The effectiveness of the proposed technique is not clear. Indicate the reference number in sentences. Besides, several techniques have been proposed in past studies. You must justify the effectiveness of the proposed technique by comparing with the latest techniques.

Answer: I've listed the references. Pls see the revised manuscript. The effectiveness of the proposed technique is achieving better output voltage with lower THD. Voltage waveform quality is a key performance index of inverter. The mandatory requirement for THD of inverter voltage is no less than 5% under any load. The lower THD means the Lower electromagnetic interference. Repetitive control and proportional-resonant control are used in inverter application, which are more concerned with the steady-state current waveforms. So these two control mainly be applied in grid-connected inverter. Standalone inverter care both steady-state and dynamic-state voltage waveform under various loads. I think the main competitor of deadbeat control is PI control, which is still the mainstream control in inverter application. PI controller is designed in continuous domain and then implemented in discrete domain by MCU chip, which is a bit embarrassing.

3, Don’t use acronym, such as PWM, THD, ZOH, and EMF, without explanation.

Answer: I have revised it. Pls see the revised manuscript.

4, In the introduction part, the research survey is thin. The existing techniques discussed in Sect. 1 is out of date. (The authors discussed the techniques proposed more than 10 years ago.) In-depth research survey is necessary. Due to poor research survey, readers will no be able to understand the problem definition of this work.

Answer: I have made some supplementary and revision. Pls see the new manuscript.

5, In the Introduction part, strong points of this proposed method should be further stated and organization of this whole paper is supposed to be provided in the end.

Answer: I have done it. Pls see the revised manuscript.

6, To help readers’ understanding, you should add a notation list.

Answer: I have added that. Pls see the revised manuscript.

7, Unify the expression, such as Gi(z) and GI(z), in figures and equations.

Answer: I have amended it. Pls see the revised manuscript.

8, In Sect.3, the simulation and experimental conditions are not clear. What kind of transistors did you use?

Answer: Transistor is IGBT made by Fairchild and model number is FGH60N60.

9, In Sect. 3, the authors discussed only the THD characteristics. Other important electrical characteristics, such as power efficiency and power factor, are not clear. Demonstrate the characteristics of the inverter clearly.

Answer: I have made corresponding supplementary. The goal of this paper is aim at achieve better inverter output voltage with low THD under various loads. So, power efficiency is not the main concern and at full load, 96% efficiency can be achieved. Power factor of standalone inverter is load dependent.

10, Future works as an integral part should be included in the Conclusions.

Answer: I have made corresponding supplementary. I think in the field of high-power inverters, the proposed scheme is worth trying. In the PWM switching frequency lower than 3-4 kHz, it’s difficult to achieve good quality voltage waveform. In the limited control bandwidth due to low switching frequency, PID control normally cannot achieve good performance and control delay is a limiting factor to reach high performance.

Round 2

Reviewer 1 Report

The review comments have been addressed in the revised version.

Reviewer 2 Report

In this paper, the authors proposed a digital control scheme for regulation of single-phase voltage source PWM (pulse width modulation) inverters, where two deadbeat controllers were used to achieve low THD (total-harmonic-distortion). In the first version, due to thin research survey and the lack of comparison between the proposed technique and the latest techniques, the effectiveness of this work was not clear. However, in the revised version, these weak points were significantly improved. The revised version is well written and organized paper, I think. It is scientifically sound and contains sufficient interest to merit publication.